# Human Male Genital Tract Microbiota

**DOI:** 10.3390/ijms24086939

**Published:** 2023-04-08

**Authors:** Arnaud Zuber, Adriana Peric, Nicola Pluchino, David Baud, Milos Stojanov

**Affiliations:** 1Materno-fetal and Obstetrics Research Unit, Department Woman-Mother-Child, Lausanne University Hospital, 1011 Lausanne, Switzerland; 2360° Fertility Center Zurich, 8702 Zollikon, Switzerland; 3Fertility Medicine and Gynaecological Endocrinology Unit, Department Woman-Mother-Child, Lausanne University Hospital, 1011 Lausanne, Switzerland; 4Faculty of Biology and Medicine, University of Lausanne, 1011 Lausanne, Switzerland

**Keywords:** male genital tract, microbiota, bacteria, infertility, sperm, prostate

## Abstract

The human body is vastly colonised by microorganisms, whose impact on health is increasingly recognised. The human genital tract hosts a diverse microbiota, and an increasing number of studies on the male genital tract microbiota suggest that bacteria have a role in male infertility and pathological conditions, such as prostate cancer. Nevertheless, this research field remains understudied. The study of bacterial colonisation of the male genital tract is highly impacted by the invasive nature of sampling and the low abundance of the microbiota. Therefore, most studies relied on the analysis of semen microbiota to describe the colonisation of the male genital tract (MGT), which was thought to be sterile. The aim of this narrative review is to present the results of studies that used next-generation sequencing (NGS) to profile the bacterial colonisation patterns of different male genital tract anatomical compartments and critically highlight their findings and their weaknesses. Moreover, we identified potential research axes that may be crucial for our understanding of the male genital tract microbiota and its impact on male infertility and pathophysiology.

## 1. Genital Microbiota

Mammal colonising microbes share a long history of coevolution with their hosts. This is the case for bacteria colonising the gut of modern primates, including humans, which arose from ancient bacteria that coevolved with the common ancestors of the lineage [1]. Gut microbiota remains the most studied bacterial community in the human body [2,3]. Despite important advances in the field, we are only starting to appreciate the impact of bacteria in the digestive tract on physiology, ranging from immunological and metabolic roles to unexpected neurobehavioral implications [4]. Today we know that most parts of the human body are colonised by a plethora of bacteria, which may greatly influence the homeostasis of these particular and very diverse ecological niches [4,5,6]. 

The genital tract is not an exception, and an increasing number of studies exploring the role of bacteria on pregnancy, infertility, and infection are being carried out [7,8,9]. Nevertheless, the majority of studies focus on the female tract, with the vaginal microbiota being the most studied environment. In a healthy state, vaginal microbiota is dominated by members of the *Lactobacillus* genus, which greatly influence this environment [10]. Recently, the existence of a specific microbiota is also being recognised in the female upper genital tract, including the uterus, fallopian tubes, and ovaries [11], which were previously considered “sterile”. In contrast, MGT microbiota has been completely neglected except in recent years, with a limited number of studies addressing the composition of bacterial communities of this particular niche [12,13,14,15]. 

The human male reproductive system sustains the production of spermatozoa and their transfer into the female genital tract for reproductive purposes. Due to its anatomical conformation, sampling of the MGT is highly invasive. Availability of samples is therefore restricted to pathological conditions such as prostate cancer, a requirement for orchiectomy or testicular biopsies in the case of infertility. On the other side, semen does not involve major restrictions for sampling and may be used as a proxy for studying the bacterial colonisation of the entire MGT.

## 2. Initial Studies on Bacteria Colonising the MGT

The presence of bacteria in the MGT has been initially associated with an infective state. The most common outcomes of bacterial infections of the MGT comprise orchitis, epididymitis, prostatitis, and urethritis. The majority of these infections are caused by sexually transmitted pathogens and ascending uropathogens. *Chlamydia trachomatis*, the most common sexually transmitted disease, and *Neisseria gonorrhoeae,* are predominant in epididymo-orchitis and urethritis [16]. On the other hand, acute and chronic prostatitis are mainly caused by *Escherichia coli*, along with other Enterobacteriaceae (*Klebsiella* spp., *Proteus* spp., and *Pseudomonas aeruginosa*), *Enterococcus* spp., and *Staphylococcus aureus* [17,18].

Bacteriospermia (presence of bacteria in the semen) was therefore linked to infections of the MGT. Bacteriological analysis of semen using conventional microbiology techniques and growth media magnified the influence and importance of known pathogenic bacteria to the detriment of nonculturable bacteria. The first studies on bacteria present in semen relied mainly on classical microbiology methods. Clinical samples were inoculated on different solid media under aerobic or anaerobic conditions to isolate bacteria. Members of *Staphylococcus*, *Enterococcus*, *Escherichia,* and *Ureaplasma* genera were the most isolated bacteria from human semen [19]. In general, these studies led to the notion that the presence of bacteria in the semen was associated with a pathological condition and that the semen of normospermic men was sterile. More recently, however, the application of polymerase chain reaction (PCR)-based methods has highlighted that bacterial DNA is present in almost all semen samples, even when microbiological investigation using conventional methods revealed the absence of bacteria [20,21]. Further characterisation of bacteria colonising human semen has been performed recently using mass spectrometry-related methods (matrix-assisted laser desorption ionisation time of flight mass spectrometry, MALDI-TOF MS) [22]. Although this technique provides a quick way to identify microorganisms, the potential presence of unidentified bacterial species could be a limitation. It is, therefore, only with the advent of NGS techniques that we started to appreciate the extent of bacterial colonisation of the MGT fully.

## 3. Microbiota of the MGT

Compared to the female counterpart, metagenomic characterisation of the MGT is still in its initial phases. Except for semen and penis, collection of samples of the MGT is highly invasive and therefore difficult to perform, also from the ethical perspective. Most of the studies performed so far rely on patients with pathologies such as prostate cancer or infertility. While paramount questions remain to be answered, such as the stability of the MGT microbiota over time or possible differences due to geographical or genetic background or hormonal imbalance, the increasing number of studies suggest that it is typically a low bacterial abundance ecosystem with relatively diverse bacterial communities. Figure 1 depicts the major genera identified with NGS in the different parts of the MGT. Further efforts should be made to understand the effect of the bacterial colonisation of the MGT for several reasons, which include a possible link with male infertility issues, its role in sexually transmitted diseases and its impact on the bacterial colonisation of the female genital tract, whose role in gynaecological and obstetrical outcomes has been established. 

### 3.1. Testes, Epididymis and Vas Deferens

Testes ensure the maintenance of the self-renewing stem cell reserve and host spermatogenesis. Replication of spermatogonia and differentiation into spermatozoa occurs in the seminiferous tubules, which constitute most of the testis’s content. The maturation of spermatozoa takes place in the epididymis, where motility and fertilising ability are acquired and where spermatozoa are stored prior to ejaculation [23].

The presence of bacteria in the upper genital tract has been associated with active infections, viral or bacterial, with subsequent acute or chronic inflammation [24]. The main bacterial agents involved may be sexually transmitted (*Chlamydia trachomatis*, *Neisseria gonorrhoeae,* and *Mycoplasma genitalium*) or associated with urinary tract infections (*Escherichia coli*, *Klebsiella pneumoniae* as *Staphylococcus aureus,* among others) and may lead to orchitis, epididymitis, or epididymo-orchitis [25]. Nevertheless, recent studies have suggested that testes without apparent signs of infection and inflammation harbour a low abundant microbiota. In a first assessment of testicular microbiota, Alfano et al. compared bacterial colonisation of men with idiopathic nonobstructive azoospermia and normal germline maturation undergoing unilateral orchiectomy for nonmetastatic seminoma [15]. As a negative control, the authors included the PC3 cell line that was grown in the presence of antibiotics, while the buccal mucosa samples were used as a positive control. Characterisation of microbiota was carried out at the phylum and class levels and revealed that Firmicutes and Actinobacteria had the highest relative abundance. At the phylum level, men with azoospermia showed a significant increase in Actinobacteria abundance, while a decrease was observed for Proteobacteria and Bacteroidetes. A more recent study questioned the existence of a testicular microbiota, with rigorous approaches concerning possible contaminations [26]. A series of negative controls and stringent in silico elimination of possible contaminants allowed the identification of specific bacterial genera specific to the testicular milieu. More in detail, these genera include *Blautia*, *Cellulosibacter*, *Clostridium XIVa*, *Clostridium XIVb*, *Clostridium XVIII*, *Collinsella*, *Prevotella*, *Prolixibacter*, *Robinsoniella*, and *Wandonia*. Interestingly, the *Prevotella* genus was described as one of the major components of seminal microbiota [13,27]. The putative impact of testicular bacteria on seminal microbiota composition may be evaluated by studying semen samples from men undergoing vasectomy. This is a common procedure aiming at definitive male contraception and eliminates the participation of testis and epididymis in the composition of semen. A pilot study in this context compared the seminal microbiota of men undergoing vasectomy prior to and after the surgery [28]. The authors concluded that in both paired and unpaired semen samples, vasectomy resulted in a decrease in α-diversity. Nevertheless, the bacterial composition of the samples (β-diversity) was not significantly different between the two groups. 

### 3.2. Accessory Glands

Most of the seminal fluid is composed of by secretions of accessory glands, which comprise the prostate, the seminal vesicles, and the bulbourethral glands. There is increasing interest in the prostate microbiota due to its potential relationship with prostate cancer [29]. Therefore, despite the invasive procedure, the availability of clinical samples is possible due to the high number of prostatectomies that are performed since prostate cancer is among the most frequently diagnosed cancers among men [30]. The prostate is one of the major accessory glands of the male reproductive tract [31]. Approximately a quarter of the ejaculate is composed of secretions produced by the prostatic epithelium [31]. While previous studies suggested the existence of a prostate microbiota [32,33], only studies relying on NGS could reveal an unbiased composition of the bacterial taxa (Table 1).

An initial analysis of the prostate microbiota of patients with an aggressive form of prostate cancer that had undergone radical prostatectomy showed that an unidentified member of the *Enterobacteriaceae* family was highly prevalent (37.2 to 81.2% of the total reads). The dominance of the Proteobacteria phylum was further denoted by the presence of *Escherichia* spp. in all samples (20.9% of the total reads). Other genera identified consistently in most of the samples comprised *Actinobacteria*, *Pseudomonas,* and *Streptococcus*, although at very low abundance.

Cavarretta et al. analysed the bacterial colonisation patterns of prostatic biopsies from patients undergoing radical prostatectomy [34]. They divided the samples into three categories, which included tumoral tissue, peritumoral tissue, and nontumoral tissue. Of note, DNA was extracted from samples embedded in paraffin, previously fixed in formalin. Among all samples, Actinobacteria was the dominating phylum, followed by Firmicutes and Proteobacteria. Major genera included *Propionibacterium*, *Corynebacterium,* and *Staphylococcus*, which comprised between 60% and 82% of all bacteria, depending on the sample group. Interestingly, *Prevotella* spp. and, more generally, members of the Bacteroidetes phylum were not detected, while this taxon was highly prevalent in other studies. 

Colonisation by members of the Proteobacteria phylum was predominant (~50%) in samples from patients with benign prostate hyperplasia [37]. Viable bacteria could be isolated from the prostatic tissue, although the colonisation patterns did not match those determined by NGS due to the limitations of laboratory cultivation of nonconventional bacterial isolates. An analogous study revealed that Firmicutes dominate in samples of patients with benign prostate hyperplasia [39]. Evidence of bacterial colonisation of the prostate was demonstrated by performing fluorescent in situ hybridisation in accordance with a low bacterial biomass microbiota. Concomitant analysis of urine samples showed that prostate microbiota was significantly different, further suggesting the existence of a local microbiota.

While several hits suggest the presence of the prostate microbiota, additional studies are required to characterise it fully. Unfortunately, the two studies about benign prostate hyperplasia [37,39] did not present the resolution of the bacterial colonisation patterns up to the genus level, making their comparison difficult. On the other side, the major differences observed at the phylum level may reflect differences due to sample and NGS processing. 

Little is known about the bacterial colonisation of the seminal vesicles and the bulbourethral glands, whose samplings remain very challenging and performed only in patients with pathologies. In a pioneer study, Lei et al. analysed bacterial colonisation profiles of samples obtained through transurethral seminal vesiculoscopy in patients with seminal vesicle complications [40]. At the phylum level, Firmicutes were the predominant taxon (52.08%), followed by Bacteroidetes (21.69%), Proteobacteria (12.72%), Actinobacteria (9.64%), and Fusobacteria (1.62%). Bacterial genera with over 5% relative abundance were *Bacteroides* (9.13%), *Lactobacillus* (5.38%), *Bifidobacterium* (5.35%), and *Faecalibacterium* (5.10%). Colonisation patterns were not significantly different between samples from patients with and without signs of infection. Nevertheless, colonisation profiles of the main bacterial genera of samples obtained from seminal vesicles did not differ significantly from urine samples obtained prior to vesiculoscopy, thus highlighting the serious risk of sample contamination during the procedure.

### 3.3. Colonisation of the Urethra

The urethra is the common duct for the evacuation of urine and the transition of semen throughout the penis. Several studies have analysed the bacterial content of the urethra by analysing first void urine samples [41,42,43]. Nevertheless, it is still not clear whether the observed taxa were present specifically in the urethra or are part of the bladder microbiota, which has not been fully elucidated yet [44]. Microbiota of first-catch urine and urethral swabs have been compared by Dong et al. [41]. Swabbing of the urethra is routinely performed in testing for sexually transmitted infections but creates discomfort for the patients. First-catch urine showed a similar microbiota compared to the swab samples. The most recurrent genera were *Lactobacillus*, *Streptococcus*, *Sneathia*, *Veillonella*, *Corynebacterium,* and *Prevotella*. In addition, the highest abundant genus in subjects with a confirmed sexually transmitted infection was *Neisseria*. Hrbacek et al. sampled first-catch urine, mid-stream urine, and aseptically catheterised urine and showed that the microbial community structure of the latter was significantly different compared to the first two [42]. This finding indirectly implies the existence of a specific urethral microbiota since the alpha diversity of first-catch and mid-stream urine samples was significantly higher. Relative abundance analyses showed that *Prevotella*_1, *Streptococcus,* and *Campylobacter* genera were specifically enriched in the first-catch urine samples and may represent specific taxa present in the urethra. Evidence of a specific urethral microbiota was also suggested by Nelson et al. by comparing voided urine specimens with swabs of the coronal sulcus [45]. 

Bacterial colonisation of the urethra has also been studied in the context of idiopathic urethritis [43]. The results suggested that microbiota is significantly different between controls and men with urethritis and that the sex of the partner also influenced the composition of the microbiota. *H. influenzae* was significantly increased in men with male partners, while *Corynebacterium* spp. Was significantly increased in men with female partners. 

### 3.4. Penis Glans and Coronary Sulcus

The penis acts as an erectile penetrating tool during sexual intercourse and allows the introduction of semen into the vagina. It is composed of several distinct parts with different physical and immunological properties. Sampling is relatively easy to perform and does not represent particular issues for the patients. Penile skin microbiota has been analysed on several occasions [46,47,48,49] (Table 2). The results showed similar colonisation patterns, dominated by *Corynebacterium* and *Staphylococcus* genera, which are typical commensals of the skin microbiota [50,51]. Different microbiota studies have focused on the foreskin mucosa and penis glans, which are the entry sites of sexually transmitted viral diseases such as human immunodeficiency virus or human papillomavirus [12]. Liu et al. identified that dysbiosis, driven by an increase in anaerobic bacteria, augmented the risk of HIV seroconversion [52]. Additional studies have suggested that microbiota dominated by anaerobic bacteria influence the local production of inflammatory chemokines that modulate the human immune system, which is the target of HIV infection [53]. More specifically, species belonging to *Peptostreptococcus*, *Prevotella,* and *Dialister* genera increased cytokine production, which resulted in the attraction of HIV-susceptible CD4+ T cells to the inner foreskin and was linked to an increased risk of HIV acquisition [54].

Penile microbiota may be drastically impacted by the circumcision procedure [55]. Increased exposition to aerobic conditions significantly reduced the abundance of putative anaerobic genera, such as *Prevotella*, *Anaerococcus*, *Finegoldia,* and *Peptoniphilus*. This was accompanied by a reduction in bacterial load and diversity, resulting in an increase in *Corynebacterium* spp. and *Staphylococcus* spp., which, as previously stated, are members of the skin microbiota. Interestingly, circumcision was also linked to a decrease in bacterial vaginosis in female partners [60]. Therefore, it has been suggested that the penile microbiota, mainly anaerobic bacteria, may trigger the activation of the immune system leading to a surge of susceptibility to sexually transmitted viral infections [61].

### 3.5. Semen Microbiota

Semen is a complex biological fluid that may be used as a proxy to study the colonisation of the MGT. Sampling is not invasive and it is routinely performed for the assessment of fertility status. Most of the studies are seminal microbiota focused, therefore, on male partners of infertile couples; the principal study question is the putative association of seminal bacteria with semen parameters (total spermatozoa count and concentration, total and progressive motility and morphology). Since the infertility factor may involve the female partner, such cohorts comprise fertile male partners with normal spermiogram parameters that can be used as internal controls for men with abnormal spermiogram parameters. In some cases, healthy sperm donors may be used as the control group, although this procedure may be limited due to ethical concerns. Alternatively, seminal microbiota was correlated to a pathological state, such as viral infections (human immunodeficiency virus or human papillomavirus) [52,62] or prostatitis [63]. Table 3 depicts all the NGS studies performed on human semen and summarises the major findings.

Almost one decade ago, Hou and colleagues performed the first metagenomic study on semen microbiota [64]. They concluded that bacteria could be identified in both fertile and infertile men and corroborated their NGS results with microscopy observation of semen samples processed with Gram staining. The most abundant genera included *Ralstonia*, *Lactobacillus*, *Corynebacterium*, *Streptococcus,* and *Staphylococcus*. *Anaerococcus* spp., the eighth most abundant genus, could be linked with negative sperm quality. 

Subsequent studies have further characterised seminal microbiota and identified specific colonisation patterns. Weng et al. identified three seminal bacterial community types that they termed G1 (*Pseudomonas*-predominant group), G2 (*Lactobacillus*-predominant group), and G3 (*Prevotella*-predominant group) in a cohort of 96 patients. *Lactobacillus* dominance was positively associated with sperm quality, while the opposite was seen with samples in which *Prevotella* spp. was dominant. Moreover, the presence of *G. vaginalis*, previously associated with bacterial vaginosis [74], was positively correlated with sperm quality. Similarly, we have previously reported three main bacterial colonisation profiles in a similar cohort of infertile couples, in two of which *Prevotella* spp. and *Lactobacillus* spp. were the predominant taxa [13]. Again, a higher proportion of semen samples of patients with abnormal spermiogram and low motility was significantly enriched with *Prevotella* spp. On the other hand, an increased abundance of *Lactobacillus* spp. positively correlated with normal semen morphology, while the presence of *Staphylococcus* spp. was linked to a normal spermiogram and high total motility. This was also the first study to quantify the bacterial load in semen using a panbacterial quantitative PCR assay, which showed that most samples carried between 10^4^ and 10^6^ copies of 16S rRNA genes per ml of semen, thus confirming the low bacterial abundance nature of this sample. Additional studies revealed the presence of *Lactobacillus*, *Prevotella*, *Staphylococcus,* and *Corynebacterium* genera in semen [68,70,72].

Nevertheless, other studies did not show the same agreement. Monteiro and colleagues analysed a seminal microbiota of 118 samples from two fertility centres in Portugal (89 patients and 29 controls). The authors found that pathogenic bacteria (*Neisseria* spp., *Klebsiella* spp., and *Pseudomonas* spp.) were associated with seminal hyperviscosity and oligoasthenoteratozoospermia. They observed a high prevalence of *Enterococcus* spp., which represented approximately 25% of total sequences. Other relatively abundant genera included *Staphylococcus*, *Anaerococcus*, *Corynebacterium*, *Peptoniphilus*, and *Propionibacterium*. The overall absence of *Lactobacillus* spp. and *Prevotella* spp. in semen samples was also observed by Lundy et al., which found *Gardnerella*, *Veilonella*, *Enterococcus*, *Streptococcus,* and *Anaerococcus* to be the most represented genera [71]. In their analysis, increased relative abundance of *Aerococcus*, *Rhodocytophaga,* and *Gemella* genera was increased in infertile patients, while *Colinsella* spp. was associated with normospermic men. 

In addition to differences in the composition of the microbiota, positive or negative associations of semen parameters with relative abundances of specific bacterial genera also showed several discordances. This was, for example, the case for *Lactobacillus* spp. [13,27,68] and *Staphylococcus* spp. [13,73], suggesting that the resolution of taxa at the genus level may not be sufficient to fully appreciate the possible impact of seminal bacteria on sperm parameters.

The different outcomes observed between studies of bacterial colonisation of semen and, more generally, of the male genital tract may be explained by multiple factors. Microbiota itself may not be stable over time, leading to different colonisation patterns being observed. Therefore, longitudinal studies should be performed to understand the dynamics of this microbiota better. Geographical variability may also be a factor, as studies have been conducted on patients from different continents. This could potentially affect the composition of the microbiota, leading to different results. 

Processing of samples is another important factor that can impact the results. Sample collection methods, DNA extraction techniques, and the presence of contaminants can all affect the composition of the microbiota that is observed. Furthermore, analysis of different variable regions of the 16S rRNA genes can also impact the final results. Finally, the bioinformatic pipeline used to analyse the data can also affect the results. Different approaches to data analysis can lead to different conclusions about the composition and function of the microbiota. Ideally, there should be standardisation of the techniques used to process samples and analyse data in order to minimise variability and ensure comparability between studies. This would improve the overall quality of research in this field and increase the reliability of findings.

## 4. How May Genital Microbiota Impair Male Fertility?

Spermatogenesis is a constant process that generates millions of spermatozoa each day in the testes [75]. It takes approximately 30–40 days for spermatogonia to undergo mitosis, meiosis, and morphological changes that will result in highly specialised cells whose goal is to fertilise the oocyte [75]. Nevertheless, spermatozoa are not functional and require a maturation stage that takes place in the epididymis and involves progressive motility [23,76]. This is also the place where motile spermatozoa are stored until ejaculation or reabsorbed in the absence of ejaculation. It is, therefore, in testes and epididymis that spermatozoa may have a higher chance of being impacted by resident microbiota. However, as seen above, studies on testicular and epididymal microbiota are scarce and warrant further analysis. On the other hand, microbiota may indirectly impact spermatozoa physiology by changing the properties of the seminal fluid. Therefore, a potential dysbiosis in the accessory glands may change the composition of prostatic or seminal vesicle fluids, thus providing a hostile environment which would not support the function of spermatozoa.

For several decades, the presence of bacteria in semen was linked to infection status and, therefore, to a reduction in spermatozoa count and an increase in leukocytes detected in sperm [77]. The adverse effect of several pathogenic bacteria on sperm physiology has been shown in the past, including *Mycoplasma genitalium*, *Mycoplasma hominis* [78], *Ureaplasma urealyticum* [79], *Chlamydia trachomatis* [80,81], and *Chlamydia*-like bacteria [82]. Direct exposure to these pathogens leads to decreased spermatozoa motility [79,83] or increased apoptosis [80,84]. Moreover, direct interaction with spermatozoa could be observed and was negatively linked with spermatozoa physiology [79,82,85,86,87]. 

More generally, bacteria may release soluble factors, such as lipopolysaccharides (LPS) [88,89], hemolysins [90,91], and other soluble spermatotoxic factors that can affect sperm physiology [92,93]. The outcomes of bacterial infection may be multiple, comprising reduction in motility, induction of teratozoospermia (abnormal sperm morphology), apoptosis, DNA fragmentation, sperm agglutination, and exposure to oxidative stress through the formation of reactive oxygen species (ROS) [94,95]. Bacteria in seminal fluid seem to trigger a local immune reaction, usually inducing leukocytospermia and cytokine secretion and leading to inflammation [96,97,98].

Other than producing cytotoxic effectors, bacteria may significantly influence their ecological niches through their metabolic activity [99,100]. A striking example is the human vagina, whose pH is acidic due to the activity of lactobacilli in a healthy state and protects against vaginal infections [101,102]. The effect of MGT microbiota on metabolites present in semen is still unknown. Metabolomic analysis of semen samples in a 660-men Chinese cohort revealed several metabolites that may represent biomarkers for the discrimination of high-quality and low-quality semen samples [103]. For example, increased levels of carnitine and its derivatives were negatively associated with semen quality. It is well established that bacteria may interfere with this metabolite and have an impact on the homeostasis of the host [104,105]. Further analyses combining metagenomics and metabolomics may unravel the potential effects of microbiota on the composition of seminal fluid and therefore explain the impact on fertility.

## 5. Future Directions and Missing Gaps

The increasing number of studies of the bacterial colonisation of the MGT, along with continuous improvements in sequencing techniques, will facilitate our understanding of MGT microbiota and its potential impact on male fertility, sexual dysfunction, and MGT pathologies, such as prostate cancer. This may also bring novel insights into the field of sexually transmitted diseases from both viral and bacterial origin. Actual discrepancies between studies may be explained by several variables, including geographical and genetic differences between cohorts, but also technical issues due to different protocols used for DNA extraction and performing of NGS. Although challenging from a technical standpoint, more research should be devoted to investigating the role of the urinary tract in the bacterial colonisation of the MGT. While there is some overlap in the bacterial genera found in both the urinary and genital tracts [106], it is crucial to determine the specific niches in which different genital bacteria reside.

One of the most important conclusions from the characterisation of MGT microbiota is that bacterial abundance is highly variable. Bacterial abundance in semen, for example, may vary from 10^8^ 16S rRNA copies per ml of semen to a virtually sterile condition [13]. Nevertheless, most semen samples contain a relatively low number of bacteria. Other body sites with these characteristics include the lower respiratory tract or the female upper genital tract [11,107]. Clinical samples with a low biomass microbiota are, therefore, very prone to contaminations, which may occur at several steps: (1) during the sampling procedure, (2) during the DNA extraction procedure, and (3) during the amplification steps required for the NGS process [108,109]. In order to accurately profile microbial communities, the inclusion of negative controls is crucial. Negative controls serve as a baseline for detecting and correcting for potential sources of contamination throughout sampling, sample processing, and data analysis. During DNA extraction, ultrapure water should be processed to ensure that the extraction process does not introduce any unwanted contaminants. Furthermore, negative controls should be included during the amplification of target DNA by PCR to account for any potential amplification of contaminations that may have occurred during this step. In this context, most of the studies on semen microbiota (Table 1) did not include negative controls in the NGS analysis, for example.

In addition to these procedures, it is also important to include in silico decontamination approaches that will use negative controls as input and filter data postanalysis, which may influence the analysis and therefore make wrong conclusions [110,111]. Moreover, an ideal pipeline used for microbial analysis should also include commercially available artificial bacterial communities, which can be used to compare the analysis outcome with their theoretical composition. Therefore, studies such as the one from Molina et al. should be taken as an example for the analysis of possible contaminations [26]. In addition, the determination of the bacterial load should be assessed using quantitative PCR.

Human ejaculate consists of a heterogeneous pool of sperm, varying in features such as shape, size, and motility, that affect the process of fertilisation. Most of the mechanisms involved in the production of this heterogenous pool are only partially known, as well as conditions affecting the dynamic changes of sperm features. Moreover, selecting the optimal population of spermatozoa is a crucial step in the ART process, and there is general agreement that the quality of sperm selection must be improved in order to optimise oocyte fertilisation. MGT microbiota analysis represents a new tool to disentangle the complexity of human sperm as well as a potential target to improve male fertility.

Future studies should evaluate the stability of the bacterial populations of the MGT, by performing longitudinal surveys of the microbiota. The presence of a dynamic and diverse microbiota may further complicate our understanding of its impact on the host. Analysis of normospermic men will be important to assess the physiological seminal microbiota. Additional strategies may be used to confirm the presence of specific bacteria in the MGT, such as FISH or electron microscopy. Furthermore, the analysis of the metabolomic impact of bacterial colonisation will be paramount to understanding the broad impact of the microbiota on the MGT and also on the female genital tract. Despite all the possible limitations, the impact of specific bacterial taxa on spermatozoa physiology should be assessed using in vitro infection models. With this regard, it will be of great importance to establish collections of genital bacteria, and more specifically, MGT bacteria, and make them available to the scientific community for further studies.

## Figures and Tables

**Figure 1 ijms-24-06939-f001:**
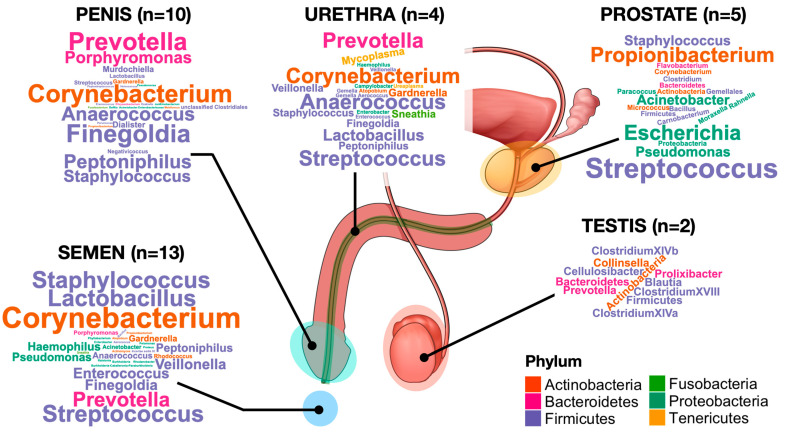
Word cloud representation of the major bacterial taxa identified in the MGT. The size of each taxon is proportional to its occurrence in all the NGS studies used to characterise the microbiota of specific parts of the MGT. The number of studies used to generate the graphs is indicated in brackets.

**Table 1 ijms-24-06939-t001:** Characterisation of prostatic tissue microbial communities using NGS.

Study	Sequencing Method	Sample Size	Negative Controls	Bacterial Count	Main Findings
Cavarretta et al., 2017 [34]	Pyrosequencing (V3–V5)	16	No	No	Three distinct areas of the prostate were analysed: tumoral, peritumoral, and nontumoral. Actinobacteria was the dominant phylum in all types of samples, followed by Firmicutes and Proteobacteria. The predominant genus was *Propionibacterium*, followed by *Corynebacterium* and *Staphylococcus*. *Staphylococcus* spp. were significantly enriched in the tumoral tissue, while *Streptococcus* spp. were almost exclusively found in the nontumoral tissue.
Yow et al., 2017 [35]	Illumina (V2–V3 and V4)	10	Yes	No	Characterisation of microbial communities in prostate tissue of men with aggressive prostate cancer. Two types of samples from the same patient: “malignant” or “benign”. Major taxa: *Enterobacteriaceae* (55.4% of total reads), *Escherichia* spp. (20.9%), *Propionibacterium acnes* (1.1%).
Feng et al., 2019 [36]	Illumina (shotgun)	65	No	No	Matched tumours and benign tissue from prostatectomy samples. Microbiota was studied using metagenomic and metatranscriptomic approaches. *Escherichia*, *Propionibacterium*, *Acinetobacter*, and *Pseudomonas* genera formed the core of the prostate microbiota. Microbiota composition did not differ significantly between tumours and the matched benign samples. Strong correlation between *Pseudomonas* expressed genes and human small RNAs that may be linked to cancer progression.
Jain et al., 2020 [37]	Illumina (V3)	20	No	No	Proteobacteria, Firmicutes, and Actinobacteria were the most common phyla. The study included bacterial culturing. Bacterial isolates identified by culturing matched with the NGS data. Samples with high (>50%) abundance of Proteobacteria showed a significative increase in DNA damage (p-γ-H2AX positive cells).
Wu et al., 2020 [38]	Illumina (V4)	63	No	No	Analysis of expressed prostatic secretions and urethral secretions from 33 patients with chronic prostatitis and 30 healthy controls. Microbial composition of expressed prostatic secretions and urethral secretions was significantly different. The most common genera were *Corynebacterium* spp., *Staphylococcus* spp., *Streptococcus* spp., *Acinetobacter* spp., and *Pseudomonas* spp. Relative abundances of *Veillonella* spp., *Atopobium* spp., and *Gemella* spp. were increased in men with chronic prostatitis compared to control group.

**Table 2 ijms-24-06939-t002:** Characterisation of penile microbial communities using NGS.

Study	Sequencing Method	Sample Size	Negative Controls	Bacterial Count	Main Findings
Price et al., 2010 [55]	Pyrosequencing (V3–V4)	12	No	No	Coronal sulcus microbiota assessment before and after circumcision. Most abundant families were *Pseudomonadaceae* and *Oxalobactericeae*, independent of the circumcision status. Significant changes in the microbiota composition after circumcision. Reduction in putative anaerobic families, especially *Clostridiales* Family XI (p = 0.006) and *Prevotellaceae*. Identification of anaerobic genera associated with bacterial vaginosis (*Anaerococcus*, *Finegoldia*, *Peptoniphilus*, and *Prevotella*).
Nelson et al., 2012 [45]	Pyrosequencing (V1–V3, V3–V5 and V6–V9)	18		No	Characterisation of coronal sulcus (CS) and urine microbiota in adolescent men. The three most abundant genera were *Corynebacterium*, *Staphylococcus*, and *Anaerococcus*, representing more than 58.9% of total sequences, followed by *Peptoniphilus*, *Prevotella*, *Finegoldia*, *Porphyromonas*, *Propionibacterium,* and *Delftia*. CS microbiota was stable during study interval, especially for *Staphyloccoccus*, *Mobiluncus*, *Prevotella*, *Dialister,* and *Anaerococcus* genera. Distinct bacterial communities between CS and urine samples and between circumcised and noncircumcised men.
Liu et al., 2013 [56]	Pyrosequencing (V3–V6)	156	No	Yes	Comparison of coronal sulcus microbiota of uncircumcised and circumcised men. Most prevalent bacterial families: *Prevotellaceae*, *Veillonellaceae*, *Clostridiales* family XI, *Actinomycetaceae*, *Coriobacteriaceae*, and *Porphyromonadaceae*. Decrease in bacterial load and diversity after 1 year of circumcision. Absolute abundance and prevalence of 12 anaerobic bacterial taxa decreased after circumcision, with a minor increase in aerobic bacteria taxa. Possible link between loss of anaerobes and effect of circumcision on HIV reduced acquisition.
Liu et al., 2015 [47]	Pyrosequencing (V3–V6)	165	No	Yes	Assessment of penile community state types (CST) and female partner’s Nugent score association. Seven profiles identified based on density (16S rRNA gene copy). Major CST groups are CST1 to 3 (low density) and CST4 to 7 (high density). Bacterial vaginosis-associated bacterial genera (*Dialister*, *Mobiluncus*, *Prevotella*, and *Porphyromonas*) more prevalent in CST4-7 compared to CST1-3. Bacteria associated with normal Nugent score (*Corynebacterium* and *Staphylococcus*) more prevalent in CST1-3 compared to CST4-7. Significant association between men with CST4 to 7 and female partners having a high Nugent score. Men with two or more extramarital partners were more likely to have CST4-7.
Liu et al., 2017 [53]	Illumina (V3–V4)	182	No	Yes	Assessment of penile anaerobe abundance and risk of HIV seroconversion. Genera with the highest risk of increased seroconversion for each 10-fold increase in abundance were *Prevotella*, followed by *Dialister*, *Peptoniphilus*, *Finegoldia*, *Porphyromonas*, *Mobiluncus*, *Peptostreptococcus*, and *Murdochiella*. Correlations between high levels of anaerobic bacteria and increased cytokine levels inducing an inflammatory response (HIV seroconversion increased risk).
Mehta et al., 2020 [57]	Illumina (V3–V4)	231	No	No	Vaginal and penile microbiota contribution to herpes simplex virus type 2 (HSV-2) serostatus within sexual partnerships. Taxa with highest mean relative abundances were *Corynebacterium* (16.4%), *Anaerococcus* (8.9%), *Streptococcus* (8.1%), *Finegoldia* (7.6%), and *L. iners* (6.8%) No microbiota differences between men with different HSV-2 statuses. Enrichment of *G. vaginalis* and *L. iners* in vagina was associated with an increased likelihood of HSV-2 in both partners. Presence of penile taxa, such as *Ureaplasma* and *Aerococcus*, linked to women HSV-2 status.
Onywera et al., 2020 [48]	Illumina (V3–V4)	238	Yes	No	Swabs of the shaft, foreskin (if uncircumcised), and glans of the penis. Most abundant bacterial families were *Corynebacteriaceae*, *Prevotellaceae*, unclassified *Clostridiales*, *Porphyromonadaceae*, *Staphylococcaceae*, *Bifidobacteriaceae,* and *Lactobacillaceae*. Six community state types (CST) identified: ○ CST-1 (most prevalent CST) dominated by *Corynebacterium.* ○ CST-2 dominated by *Corynebacterium*, unclassified *Clostridiales*, and *Porphyromonas*. ○ CST-3 dominated by *Gardnerella* and *Corynebacterium*. ○ CST-4 dominated by *Chryseobacterium*, *Corynebacterium*, and *Acinetobacter*. ○ CST-5 dominated by *Prevotella*, unclassified *Clostridiales*, *Corynebacterium*, and *Porphyromonas*. ○ CST-6 (least prevalent CST) dominated by *Lactobacillus* with very low relative abundance of *Corynebacterium*. CST-5 more likely associated with HPV or HR-HPV infections than CST-1.
Plummer et al., 2021 [49]	Illumina (V3–V4)	34	Yes	No	Analysis of microbiota by cutaneous penile swabs and first-pass urine samples before and after antibiotic treatment. At day 0, most abundant genera in the cutaneous penile microbiota were *Corynebacterium*, *Finegoldia*, *Staphylococcus*, *Peptoniphilus,* and *Prevotella*. At day 0, most abundant taxa in the urethral microbiota were *Streptococcus*, *Lactobacillus iners*, *Gardnerella*, *Sneathia,* and *Staphylococcus*. Male genital specimens differed after 7 days of antibiotic treatment.
Watchorn et al., 2021 [58]	Illumina (V3–V4)	40	No	Yes	Balanopreputial swabs and urine samples of uncircumcised patients with male genital lichen sclerosus (MGLSc). Microbiota differed significantly between healthy and men with MGLSc. No difference in the bacterial load between the two groups. In balanopreputial sac: *Finegoldia* spp. median relative abundance in MGLSc patients lower than in controls; *Fusobacterium* spp. and *Prevotella* spp. median relative abundance higher in MGLSc patients. In urine: *Finegoldia* spp. median relative abundance comparable in the two groups; *Fusobacterium* spp. abundance was higher in MGLSc patients.
Prodger et al., 2021 [54]	Pyrosequencing (V3–V6)	188	No	Yes	Analysis of the penile foreskin microbiota Major bacterial genera: *Prevotella*, *Peptoniphilus*, *Porphyromonas*, *Finegoldia*, *Corynebacterium*, *Anaerococcus*, and *Dialister.* High abundance of *Peptostreptococcus*, *Prevotella,* and *Dialister* genera was linked with an increased production of inflammatory cytokines. Increased inflammation resulted in the attraction of HIV-susceptible CD4+ T cells and was linked to an increased risk of HIV acquisition.
Mehta et al., 2020 [59]	Illumina (V3–V4)	168	Yes	No	Assessment of bacterial vaginosis incidence in women based on prediction of penile microbiota. Most prevalent taxa relative abundances such as *Corynebacterium*, *Streptococcus*, *Anaerococcus*, and *Finegoldia* were similar for meatal and glans/coronal sulcus samples, but overall composition differs. Most important taxa predicting bacterial vaginosis: *Parvimonas* spp., *Lactobacillus iners*, *Fastidiosipila* spp., *Negativicoccus* spp., *L. crispatus*, *Dialister* spp., *Sneathia sanguinegens*, *Gardnerella vaginalis*, *Prevotella corporis*, and *Corynebacterium*.

**Table 3 ijms-24-06939-t003:** Characterisation of seminal microbiota using NGS.

Study	Sequencing Method	Sample Size	Negative Controls	Bacterial Count	Main Findings
Hou et al., 2013 [64]	Pyrosequencing (V1–V2)	77	No	No	Six groupings of semen community types. Main genera: ○ I: *Streptococcus*, *Corynebacterium*, *Finegoldia,* and *Veillonella* ○ II: *Prevotella*, *Peptoniphilus*, *Lactobacillus,* and *Porphyromonas* ○ III: *Corynebacterium*, *Staphylococcus*, *Finegoldia,* and *Anaerococcus* ○ IV: *Ralstonia* ○ V: *Lactobacillus* ○ VI: *Atopobium*, *Veillonella*, *Prevotella*, *Aerococcus,* and *Gemella* No overall differences in bacterial communities between sperm donors and infertile patients. *Anaerococcus* spp. Negatively associated with sperm quality.
Weng et al., 2014 [27]	Illumina (V4)	96	No	No	Most abundant genera: *Lactobacillus*, *Pseudomonas*, *Prevotella,* and *Gardnerella*. Three main clusters of seminal bacteria: ○ *Lactobacillus* predominant group ○ *Pseudomonas* predominant group ○ *Prevotella* predominant group *Lactobacillus* and *Gardnerella* spp. specifically enriched in normospermic subjects. *Prevotella* spp. higher in low-quality samples.
Liu et al., 2014 [52]	Pyrosequencing (V3–V6)	49	No	No	*Streptococcus, Corynebacterium, and Staphylococcus* were the predominant genera, irrespective of the HIV status of the patients. *Ureaplasma* spp. specifically enriched in HIV-uninfected men. *Mycoplasma* spp. specifically enriched in HIV-infected men. Association between HIV infection and decreased semen microbiota diversity and richness. Restoration of microbiota after HIV therapy.
Mändar et al., 2015 [65]	Illumina (V6)	23	No	Yes	Seminal bacterial communities more diverse but less abundant compared to the vaginal microbiota. Predominance of *G. vaginalis* in female partners was related to inflammation of genital tracts of male partners. Decrease in *Lactobacillus crispatus* relative abundance after sexual intercourse. High concordance between semen and vaginal samples.
Mändar et al., 2017 [63]	Illumina (V6)	67	No	No	Firmicutes comprised 41.7% of all sequences. Higher lactobacilli abundance in healthy men compared to prostatitis patients. Proteobacteria enriched in prostatitis patients compared to healthy men. Species richness was higher in prostatitis patients compared to healthy men.
Chen et al., 2018 [66]	Illumina (V4)	17	No	No	Predominant genera in all samples: *Lactobacillus, Prevotella, Proteus, Pseudomonas, Veillonella, Corynebacterium, Rhodococcus, Staphylococcus, and Bacillus.* *Campylobacter* and *Plesiomonas* spp. specifically enriched in the obstructive azoospermic subjects. *Sneathia* and *Lysobacter* spp. specifically enriched in the nonobstructive azoospermic subjects.
Monteiro et al., 2018 [67]	Ion Torrent (V3–V6)	118	No	No	Predominant genera: *Enterococcus, Staphylococcus, Anaerococcus, Corynebacterium,* and *Peptoniphilus*. Partial correlation of an increase in *Neisseria*, *Klebsiella,* and *Pseudomonas* genera with seminal hyperviscosity and oligoasthenoteratozoospermia, with a concomitant reduction in *Lactobacillus* spp.
Baud et al., 2019 [13]	Illumina (V1–V2)	94	Yes	Yes	Three clusters of seminal microbiota communities: ○ *Prevotella* spp. predominant ○ *Lactobacillus* spp. predominant ○ Polymicrobial group Highest bacterial load in the *Prevotella* group. *Prevotella* associated with abnormal spermiogram parameters. *Lactobacillus* enriched in samples with normal morphology. *Staphylococcus* enriched in the normospermic group.
Yang et al., 2020 [68]	Illumina (V1–V2)	159	No	No	Different composition of seminal microbiota between patients with asthenospermia and oligoasthenospermia compared to controls. Relative abundance of *Ureaplasma*, *Bacteroides*, *Anaerococcus*, *Finegoldia*, *Lactobacillus* spp., and *Acinetobacter lwoffii* increased in subjects with asthenospermia. *Lactobacillus* spp. enriched in subjects with oligoasthenospermia.
Amato et al., 2020 [69]	Illumina (V3–V4)	23	No	Yes	No difference between seminal microbiota of men with idiopathic infertility and controls. No differences in seminal microbiota between intrauterine insemination success and failure groups.
Štšepetova et al., 2020 [70]	Pyrosequencing (NA)	50	Yes	Yes	Decreasing bacterial loads found in raw, processed, and sperm samples used for oocyte insemination. *Staphylococcus* spp. was found only in semen from patients with inflammation. Negative correlation between Bacteroidetes and sperm motility. Negative correlation between Alphaproteobacteria embryo quality.
Lundy et al., 2021 [71]	Illumina (V3–V4)	37	No	NA	Infertile men had an increased seminal alpha diversity and distinct beta diversity compared to controls. Increased seminal *Aerococcus* spp. and decreased rectal *Anaerococcus* spp. in infertile subjects. *Prevotella* spp. abundance was inversely associated with sperm concentration. *Pseudomonas* spp. was directly associated with total motile sperm count. Vasectomy altered seminal microbiota.
Tuominen et al., 2021 [62]	Illumina (V3–V4)	31	No	No	Major taxa in semen: *Comamonadaceae*, *Bifidobacteriaceae*, *Tissirellaceae*, *Corynebacteriaceae*, *Delftia*, *Propionibacterium,* and *Streptococcus*. Human papilloma virus status did not impact alpha or beta diversities. Increased relative abundance of *Moraxellaceae*, *Streptococcus*, and *Peptostreptococcus* taxa in subject with human papilloma virus.
Yao et al., 2021 [72]	Illumina (V3–V4)	87	No	No	Worse sperm parameters observed in leukocytospermia-related groups. Increased alpha diversity in the leukocytospermia-related groups. Two profiles identified: ○ *Lactobacillus*-enriched group ○ *Streptococcus*-enriched group Majority of the samples in the normal seminal leukocyte count group categorised as *Lactobacillus*-enriched. Majority of samples in the leukocytospermia group were categorised as *Streptococcus*-enriched.
Bukharin et al., 2022 [73]	Illumina (NA)	72	No	NA	*Staphylococcus* most abundant overall genus most abundant, specifically enriched in infertile subjects. Cell-free supernatants of *Staphylococcus* spp. isolated from healthy subjects reduced levels of IL-10 and IL-17 cytokines more efficiently compared to infertile subjects. *Staphylococcus* and *Enterococcus* species from infertile subjects had an increased level of biofilm formation compared to healthy subjects.

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
