# Peer review of "Human Male Genital Tract Microbiota"

_ijms, 2023, doi:10.3390/ijms24086939_

Round 1

Reviewer 1 Report

       This article reviews the results of studies using the NCS technique to explore the colonization of specific microbiota in different parts of the male genital tract, which provides a new direction for the treatment of male diseases and is innovative and recommended for acceptance, but the article still has some problems.
[1] In the review of the study using urine to investigate the specific urogenital microbiota, the authors mentioned the possible influence of the bladder microbiota on their results, however, the urinary tract also includes the kidney and ureter, whether the microbiota present in these two places also have some influence on their results, if so, it is suggested that the authors add the influence of the microbiota present in the kidney and ureter on their results in the article.
[2] In the studies reviewing the relationship between semen microbiota and male fertility, there is some variability in the results obtained from different studies, and it is suggested that the authors analyze the reasons for this variation in their article.
[3] In this article, the species and abundance of specific microbiota in different parts of the male reproductive tract are reviewed, and for a better understanding of the article, it is suggested that the authors add tables of relevant content to the article.

Author Response

We thank the reviewer for the positive feedback regarding the manuscript.

[1] In the review of the study using urine to investigate the specific urogenital microbiota, the authors mentioned the possible influence of the bladder microbiota on their results, however, the urinary tract also includes the kidney and ureter, whether the microbiota present in these two places also have some influence on their results, if so, it is suggested that the authors add the influence of the microbiota present in the kidney and ureter on their results in the article.

We agree with the reviewer’s reasoning. Nevertheless, not much is known in the field of the upper urinary tract, however. As for the semen, urine may be a proxy of the colonisation of the whole urinary tract. In this view, we suggest an addition in the last section of the manuscript, underscoring the necessity for further investigation to determine the influence of the urinary tract on the colonization of the MGT (Lines 379-383).

[2] In the studies reviewing the relationship between semen microbiota and male fertility, there is some variability in the results obtained from different studies, and it is suggested that the authors analyze the reasons for this variation in their article.

This is an interesting point: we added a section in the text concerning this issue. (Lines 310-326).

[3] In this article, the species and abundance of specific microbiota in different parts of the male reproductive tract are reviewed, and for a better understanding of the article, it is suggested that the authors add tables of relevant content to the article.

As suggested by the reviewer, we added two additional tables (Table 1 and Table 2), which describe the most studied sites, in addition to the seminal microbiota studies.

Reviewer 2 Report

Arnaud Zuber et al. updated the knowledge of human male genital tract microbiota by systematically describing the species, the range of colonization, and the possible effects of microbiota in the male genital tract. In general, this article highlight of the topic of human male genital tract microbiota. I would recommend this manuscript because it will be of wide interest to the field of reproductive medicine. However, this manuscript still needs to make some corrections before publication.

1. This manuscript is only about the human male genital tract microbiota, and the relationship with male fertility is only speculated through statistical analysis, lacking supporting evidence from relevant animal experiments, resulting in a lack of solid conclusions. I would recommend the authors change the title to “Human male genital tract microbiota” or add the results of relevant animal experiments.

2. Line 15-16: The abstract should be concise and focused on the male genital tract microbiota topic. Please delete “but most of the studies focused on the female genital tract and mostly on the vaginal niche.”

3. Line 17: “suggest that bacteria may have a role in male infertility and pathological conditions”. I think the word “may” significantly reduces the value of this article, and since there is already a “suggest”, I recommend removing “may”.

4. Line 56: The second section, Initial studies on bacteria colonising the MGT, and the third section, Microbiota of the MGT, are similar in content, and the second section is more like a summary of the third section. I suggest combining these two parts into one section.

5. Line 72-79: Normospermic men were sterile and confirmed by conventional methods. However, PCR-based methods have highlighted that bacterial DNA is present in almost all semen samples. I think that the semen of fertile people should be tested for bacteria under a microscope, and I suggest that studies related to semen bacterial examination should be expanded.

6. Line 72-79: Figure 1: I recommend using the same color for the same bacteria and one family of colors for the same family of bacteria. It is convenient to use different colors to distinguish different bacteria so that the readers can find out the difference in bacterial colonisation the first time.

7. Line 113: Please add a qualifying modifier before "testes" because the microbiota in "orchitis" may not be in low abundance.

8. The manuscript repeatedly mentions that sample contamination interferes with microbiota analysis and leads to erroneous conclusions in Line 193 and Section 5 Future directions and missing gaps but finally does not mention what can be done to strengthen the management of sample collection or what other assays can be combined to minimize errors. I suggest the authors give some valid comments in the corresponding position in the manuscript.

9. Line 231-233: I think it's hard to draw such a conclusion in a definable tone because Cindy M Liu only evaluated semen samples from 22 HIV-uninfected men who have sex with men and 27 HIV-infected men who have sex with men. Certainly, the author should also provide more information about HIV seroconversion and anaerobic bacteria, such as (JCI Insight. 2021 Apr 22;6(8):e147363. doi: 10.1172/jci.insight.147363) and (mBio. 2017 Jul 25;8(4):e00996-17. doi: 10.1128/mBio.00996-17).

10. Line 296-301: This paragraph severely diminishes the role of genital tract microbiota in male fertility. I recommend that the authors add more relevant literature and discuss the reasons for the inconsistency in detail.

11. Line 302: This manuscript was introduced by the gut microbiota. However, the main mechanisms of gut microbiota changes and male genital tract microbiota changes are very different. Gut microbiota dysbiosis more frequently is a problem with immune metabolism, while male genital tract microbiota changes are more of an invasive infection. It is recommended that the authors add in this section a comparison of the pathogenic mechanisms of genital tract microbiota and more studied flora such as gut microbiota.

Author Response

We thank the reviewer for the constructive feedback and valuable suggestions that have improved the quality of the manuscript.

  1. This manuscript is only about the human male genital tract microbiota, and the relationship with male fertility is only speculated through statistical analysis, lacking supporting evidence from relevant animal experiments, resulting in a lack of solid conclusions. I would recommend the authors change the title to “Human male genital tract microbiota” or add the results of relevant animal experiments.

We agree with the reviewer that the title can be modified, since experiments on animals are out of scope of this review. We therefore propose “Human male genital tract microbiota”.

  1. Line 15-16: The abstract should be concise and focused on the male genital tract microbiota topic. Please delete “but most of the studies focused on the female genital tract and mostly on the vaginal niche.”

We modified the abstract according to the reviewer’s comment (Lines 15-18).

  1. Line 17: “suggest that bacteria may have a role in male infertility and pathological conditions”. I think the word “may” significantly reduces the value of this article, and since there is already a “suggest”, I recommend removing “may”.

We agree with the proposition of the reviewer (Line 16).

  1. Line 56: The second section, Initial studies on bacteria colonising the MGT, and the third section, Microbiota of the MGT, are similar in content, and the second section is more like a summary of the third section. I suggest combining these two parts into one section.

We think that the actual order of the manuscript has its logic:

  • In section 2 we give a historical perspective, with the main notion that presence of bacteria in the MGT was associated with an infection rather than colonization.
  • We then focus on the colonization of the MGT by NGS in the section 3.
  • In section 4, we address the possible influence of the microbiota on infertility.

 We therefore propose to keep the actual organization of the text.

  1. Line 72-79: Normospermic men were sterile and confirmed by conventional methods. However, PCR-based methods have highlighted that bacterial DNA is present in almost all semen samples. I think that the semen of fertile people should be tested for bacteria under a microscope, and I suggest that studies related to semen bacterial examination should be expanded.

We agree with the reviewer. We added a sentence highlighting the need to analyse normospermic men in the last section (Lines 420-421). In this section we also point to the fact that analysis of semen by microscopy should be increasingly used to study bacterial colonisation is situ.

  1. Line 72-79: Figure 1: I recommend using the same color for the same bacteria and one family of colors for the same family of bacteria. It is convenient to use different colors to distinguish different bacteria so that the readers can find out the difference in bacterial colonisation the first time.

We agree with the reviewer to add additional information regarding the bacterial taxa. Nevertheless, adding a color code for each family made the figure very busy. We suggest that the classification at the phylum level may be clearer. We therefore propose the new Figure 1.

  1. Line 113: Please add a qualifying modifier before "testes" because the microbiota in "orchitis" may not be in low abundance.

We specified that a low abundance microbiota is present in “testes without apparent signs infection and inflammation” (Line 117).

  1. The manuscript repeatedly mentions that sample contamination interferes with microbiota analysis and leads to erroneous conclusions in Line 193 and Section 5 Future directions and missing gaps but finally does not mention what can be done to strengthen the management of sample collection or what other assays can be combined to minimize errors. I suggest the authors give some valid comments in the corresponding position in the manuscript.

As suggested, we added a section on what can be done to improve the quality of microbiota data. (Lines 392-399).

  1. Line 231-233: I think it's hard to draw such a conclusion in a definable tone because Cindy M Liu only evaluated semen samples from 22 HIV-uninfected men who have sex with men and 27 HIV-infected men who have sex with men. Certainly, the author should also provide more information about HIV seroconversion and anaerobic bacteria, such as (JCI Insight. 2021 Apr 22;6(8):e147363. doi: 10.1172/jci.insight.147363) and (mBio. 2017 Jul 25;8(4):e00996-17. doi: 10.1128/mBio.00996-17).

We have developed more this part with the help of the references given by the reviewer (Lines 237-242).

  1. Line 296-301: This paragraph severely diminishes the role of genital tract microbiota in male fertility. I recommend that the authors add more relevant literature and discuss the reasons for the inconsistency in detail.

We added two additional paragraphs (Lines 310-326) that consider possible reasons of the inconsistencies observed between studies.

  1. Line 302: This manuscript was introduced by the gut microbiota. However, the main mechanisms of gut microbiota changes and male genital tract microbiota changes are very different. Gut microbiota dysbiosis more frequently is a problem with immune metabolism, while male genital tract microbiota changes are more of an invasive infection. It is recommended that the authors add in this section a comparison of the pathogenic mechanisms of genital tract microbiota and more studied flora such as gut microbiota.

We acknowledge and agree with the reviewer's comment. However, we believe that the purpose of this brief introduction is not to delve into the intricacies of gut microbiota dysbiosis mechanisms, but rather to acquaint the reader with the idea that microbiota can significantly affect the host. Therefore, we believe that the current text does not require any modifications.

Author Response

We thank the reviewer for the positive feedback.

Reviewer 4 Report

Bacterial infestation of male reproductive organs and fluids has become an important player in the etiology of male sub- or infertility. In this sense, the paper represents an important summary of the bacterioflora present in male genital tract and its possible implications on the resulting fertility. The paper is well-structured and critically addresses the most important findings from relevant studies focused on the bacterial colonization profiles of different male genital structures. The paper reads well, and I acknowledge its potential to attract the attention of scientists and professionals working in the field of andrology and reproductive biology. I have only a few minor comments:

-          Whilst it is clearly stated that the review focused on NGS studies, Section 2 mentions available techniques used to analyze bacterial profiles of biological samples. I am missing mass spectrometry that has been used to identify bacterial profiles of semen. It may be worthwhile to briefly mention this technique and its strengths or weaknesses in terms of bacterial identification in andrology.

-          I would highly recommend summarizing the most important studies relevant to each section of the manuscript into tables. This may enable a faster orientation of the reader.

-          Based on the information gathered from the collected data indicate differences in the bacteriome amongst healthy males and patients objectively suffering from an ailment? Are there any studies done on the urogenital or seminal bacteriome in healthy and normozoospermic individuals?

Author Response

We thank the reviewer for the positive comments about our manuscript and for the useful suggestions. Here are our answers to the points raised by the reviewer:

  • Whilst it is clearly stated that the review focused on NGS studies, Section 2 mentions available techniques used to analyze bacterial profiles of biological samples. I am missing mass spectrometry that has been used to identify bacterial profiles of semen. It may be worthwhile to briefly mention this technique and its strengths or weaknesses in terms of bacterial identification in andrology.

We added the information about bacterial identification by mass spectrometry in the manuscript (Lines 76-80).

  • I would highly recommend summarizing the most important studies relevant to each section of the manuscript into tables. This may enable a faster orientation of the reader.

As suggested by the reviewer, we added two additional tables (Tables 1 & 2) that summarize the main findings of NGS studies focusing on prostate and penis microbiota.

  • Based on the information gathered from the collected data indicate differences in the bacteriome amongst healthy males and patients objectively suffering from an ailment? Are there any studies done on the urogenital or seminal bacteriome in healthy and normozoospermic individuals?

Regrettably, there have been no studies specifically dedicated to normospermic individuals. However, among the cohorts of infertile couples, some cases have been identified in which the infertility factor is linked to the female partner (as discussed in lines 259-262). Analysis of these samples has indicated the presence of a normal microbiota, even in the absence of any specific clinical symptoms.